# Production of MCM-41 Nanoparticles with Control of Particle Size and Structural Properties: Optimizing Operational Conditions during Scale-Up

**DOI:** 10.3390/ijms21217899

**Published:** 2020-10-24

**Authors:** Rafael R. Castillo, Lorena de la Torre, Félix García-Ochoa, Miguel Ladero, María Vallet-Regí

**Affiliations:** 1Department of Chemistry in Pharmaceutical Sciences, Faculty of Pharmacy, Universidad Complutense de Madrid, 28040 Madrid, Spain; rafcas01@ucm.es; 2Biomedical Research Centre Network (CIBER), Av. Monforte de Lemos 3-5, 28029 Madrid, Spain; 3imas12 Research Institute, Hospital 12 de Octubre, Av. Córdoba s/n, 28041 Madrid, Spain; 4Department of Chemical and Materials Engineering, Faculty of Chemistry, Universidad Complutense de Madrid, 28040 Madrid, Spain; lorenadelatorresierra@gmail.com (L.d.l.T.); fgochoa@ucm.es (F.G.-O.)

**Keywords:** MCM-41 synthesis, particle size, particle shape, operational conditions, scale-up

## Abstract

The synthesis of Mobil Composition of Matter 41 (MCM-41) mesoporous silica nanoparticles (MSNs) of controlled sizes and porous structure has been performed at laboratory and pilot plant scales. Firstly, the effects of the main operating conditions (TEOS –Tetraethyl ortosilicate– addition rate, nanoparticle maturation time, temperature, and CTAB –Cetrimonium bromide– concentration) on the synthesis at laboratory scale (1 L round-bottom flask) were studied via a Taguchi experimental design. Subsequently, a profound one-by-one study of operating conditions was permitted to upscale the process without significant particle enlargement and pore deformation. To achieve this, the temperature was set to 60 °C and the CTAB to TEOS molar ratio to 8. The final runs were performed at pilot plant scale (5 L cylindrical reactor with temperature and stirring speed control) to analyze stirring speed, type of impeller, TEOS addition rate, and nanoparticle maturation time effects, confirming results at laboratory scale. Despite slight variations on the morphology of the nanoparticles, this methodology provided MSNs with adequate sizes and porosities for biomedical applications, regardless of the reactor/scale. The process was shown to be robust and reproducible using mild synthesis conditions (2 mL⋅min^−1^ TEOS addition rate, 400 rpm stirred by a Rushton turbine, 60 min maturation time, 60 °C, 2 g⋅L^−1^ CTAB, molar ratio TEOS/CTAB = 8), providing *ca*. 13 g of prismatic short mesoporous 100–200 nm nanorods with non-connected 3 nm parallel mesopores.

## 1. Introduction

Since their discovery, mesoporous silica has been broadly employed in many fields thanks to a very convenient and regular porosity showing high specific surface areas. The application of the developed technology in recent decades has permitted to synthesize a large number of materials (Santa Barbara Amorphous 15 and 16 –SBA-15 and SBA-16–, MCM-41, MCM-48, Fibrous silica nanospheres –KCC-1–, among others) with different pore sizes and geometries (hexagonal, cubic, radial, etc.) and particle morphologies (spheres, rods, layered, etc.). Their applications are consequences of their porous nature and robust chemical composition, which permit their use in many applications: water treatment [1], gas processing [2], and supported catalysis [3], amongst many others. Nevertheless, the most prominent field of research for those materials is nanomedicine, in which Mesoporous Silica Nanoparticles (MSNs) have demonstrated their capacity as drug delivery nanocarriers [4,5,6] and new-generation hybrid nanocomposites [7]. For instance, the use of mesoporous silica MSNs was permitted to achieve relevant nanomedical advances, such as: (1) the preparation of targeted nano-systems able to discriminate between cell populations [8,9]; (2) the development of nano-systems able to modify cells’ gene expression [10,11]; (3) the combination of proapoptotic therapeutic effects [12], and (4) the creation of theragnostic nanodevices [13,14] and sensors [15].

The proliferation of silica-based nanomaterials for biomedical purposes [16,17] is consequence of its intrinsic properties that combine an outstanding loading capacity, which clearly outrange other nano-systems, with a remarkable biocompatibility [18,19,20,21,22,23], good physicochemical profile, and facile functionalization. Nevertheless, to create a therapeutic nano-systems from a nanoparticle, it is important to modify the surface to provide the desired recognition capabilities, [24] immune stealthiness [20], and, in the case of MSNs, improved release profiles [25]. However, the particles themselves must also comply with a series of parameters that ensure their suitability for in vivo applications [26]. From those intrinsic to the nanoparticle, the most discriminating factor is size, which show an optimal range within a hundred of nanometers [27,28]. This optimal value is obtained considering that smaller particles are quickly removed from bloodstream by the renal system and those bigger than 300 nm could be blocked on capillary vessels and pulmonary alveoli leading to embolisms. Thus, focusing on this size limitation, SBA-15 and MCM-48 like morphologies should be discarded as both are usually obtained exceeding the upper limit; while among those remaining, MCM-41 is preferred for drug delivery because of its ordered and non-connected pore distribution. In fact, the broader dendritic pore morphology of KCC-1 [29] MSNs is highly affected by the concentration of loaded molecules, oppositely to molecules (that can be a drug, prodrug, fluorophore, etc.) loaded in MCM-41, whose mass transport obeys a Fickian diffusion equation [30].

Because of the growing interest of MSNs in nanomedicine, it is of interest to develop a synthetic procedure that permits to prepare MCM-41 like MSNs, in relatively large scale, with a controlled size and good reproducibility. Typically, MCM-41 MSNs are crafted by a modified Stöber methodology [31]; employing a basic [32] solution of aqueous cetyltrimethylammonium halide (0.5 L of 5.5 mM) at medium temperatures (≤ 80 ºC) into which is added Si (OEt)_4_ as silica precursor. However, in spite of the procedural similarities, the particles morphologies differ from each synthesis conditions and even between batches. Unfortunately, this variability could not be justified by only a thermal or reagents effect, so a systematic study of the process must be addressed in order to have a scalable and reproducible synthetic methodology suitable for future nanotechnological developments.

Besides the logic variation of reaction rates promoted by the variation of chemical species involved in the synthesis [33,34], the preparation of MSNs is also highly affected by the methodology employed. For this reason, many research groups have reported their particular synthesis methods in order to improve the resulting particles, focusing mainly on shape, final size, and polydispersity. However, many of these modifications have to sacrifice yields in order to preserve the desired textural properties or vice versa.

For silica microparticles, but not nanoparticles, Šoltys et al. scaled up the particle production comparing a volume-based and a concentration-based procedure. The first one resulted in a 40-fold to 50-fold increase in particle production depending on the type of SiO_2_ microparticle; the agitation was the main variable influencing the particle size and, therefore, the success of the scale-up procedure. In the concentration-based scale-up, a 16-fold production increase was attained, with the conservation of the pore shape but increasing the internal pore volume and specific surface [35]. Nevertheless, among the extensive bibliography regarding synthesis of MSNs, there could be interesting contributions that analyze the effect of some variables on the synthetized product. For instance, Nooney et al. studied the dilution of reagents, finding that higher dilutions gave smaller particles [36]. Regarding pH, Huo and coworkers found that lower pH values produced bigger particles [32], while Varache et al. set the optimal concentration of NaOH for the synthesis of MCM-41 like MSNs in the range of 15 mM [37]. Additionally, according to Chiang et al., the initial optimal pH value must be comprised between 12 and 13, because beyond those limits the resulting meso-order decreases significantly [38]. Another interesting contribution, by Yamada et al., studied the ratio between surfactant and silica precursor, which must be set to 0.12 [39] for an optimal pore distribution, according these authors, although in this case for dendritic structures. In summary, despite all the available knowledge, the reality is that very few research papers deal with the influence of the main operating conditions on the properties of MSNs obtained at large scale [40,41], which gives an idea of the complexity of this process.

The aim of this work is to carry out a systematic study of the influence of the main variables and operating conditions in the properties of the MCM-41 synthetized particles, with emphasis in the control of the particle size, morphology, and other structural properties. The variables to be considered are, in a preliminary stage, the initial pH and the quality of the reagents. Then, several operating conditions will be studied using a progressive approach via a Taguchi DOE (Design of Experiments) to establish which variables have a significant effect on the product properties, followed by a one-variable-at-a-time approach to fine-tune the main variables. Finally, the results are tested in a bigger reactor (volume-based scale-up) as, to our knowledge, there is no study in the literature regarding the scale-up of MCM-41.

## 2. Results and Discussion

The synthesis of mesoporous materials is based on the hydrolytic polymerization of a SiO_2_ precursor onto a pre-formed CTAB micellar phase responsible of the final meso-order. The most common precursor for this process is TEOS because of its chemical stability (higher than the tetramethyl analogue), availability and price. To prepare the silica nanoparticles, TEOS must be hydrolyzed under basic conditions to produce silanol groups able to self-condense. In any case, regardless of the base employed, the resulting silanol groups are responsible of the TEOS condensation; although at the expense of having significant variability depending on the base strength and concentration. Hence, to avoid this variability a strong base (NaOH) was employed in this work, and the initial pH was fixed for each experiment in a value of 11.8 [37,38]. 

### 2.1. Preliminary Study: Effect Of Initial Ph And Teos Quality

During the optimization process the behavior of how different reagent’s qualities would affect the morphology and textural properties of the obtained nanoparticles was evaluated. Thus, three different experiments were conducted, in which the quality of the employed TEOS varied, employing the different qualities available at the supplier: 98% (reagent grade), 99%, and 99.999% (trace metals basis quality).

Synthetized MSNs with different TEOS qualities did not show significant differences between batches, suggesting that the reagent grade TEOS was enough to achieve satisfactory results. However, since this silica precursor is moisture sensitive, an inadequate handling and storage may undergo an uncontrolled hydrolysis that would provoke self-condensation able to affect the reaction outcome. Moreover, the aging of NaOH solution has reported to produce carbonated solutions able to affect the reaction outcome by shifting the morphology from rounded to amorphous and increasing the overall size [37]. The remainder reagent (CTAB) and the solvent (double distilled water), showed negligible impact on the morphology and textural properties of prepared MSNs.

On the other side, the effect of pH was extensively studied by Varache et al. [37], who reported an optimal NaOH concentration between 10 and 15 mM. Since our conditions employ NaOH concentration within their optimal range, we have not modified this parameter: pH was set at 11.8. 

### 2.2. Taguchi Optimization Method: Study Of Different Factors

Given that one of the goals is to scale-up the process to obtain nanoparticles with applicability in nanomedicine, 150 nm seems to be the most adequate particle size for medical application. Therefore, the signal-to-noise ratio (S/N ratio) should be the nominal-the-better, and can be estimated using Equation (1):(1)$$S/N\mathrm{ratio}\;\left(x\right)=10log\frac{\left(\bar{y^{2}}\right)}{s^{2}}\;$$
where x is a given value or level of a tested independent variable or factor, $\bar{y}$ is the average value of the dependent variable (the particle size, in this case) for such x value, and s^2^ is the variance of the y values for such x value. The average value of the particle size ($\bar{y}$) and its variance (s^2^) are computed using the method of moments for the peaks of particle size distribution as measured by TEM with OriginLab®2016. 

The Taguchi method permits to understand how an independent or operational variable or factor affects the dependent value of interest; in particular, the average S/N for each level or value of the factor being tested is given by the following equation:(2)$$\bar{S/Nratio}=\frac{\mathrm{Sum}\;\mathrm{of}\;S/N\;\mathrm{values}\;\mathrm{for}\;\mathrm{factor}\;i\;\mathrm{at}\;\mathrm{each}\;\mathrm{level}\;j}{3}$$

The S/N ratio values were calculated for each factor at all levels or values and compared to a global S/N ratio average value estimated for all analyzed factors and levels. If the S/N ratio values of a given factor varies considerably in comparison to the global average, the factor tested significantly affects the response analyzed, also indicating the trend that the response follows with the value of the factor analyzed. In case of being interested in a particular local optimum value for the tested variables, the best values for all factors are those leading to the higher S/N ratio values and can be employed to estimate an optimal value of the dependent variable. For example, for four given factors, A, B, C, D, the optimum value can be calculated as:(3)$$\mathrm{Optimum}=Y+\left(A_{\mathrm{opt}}-Y\right)+\left(B_{\mathrm{opt}}-Y\right)+\left(C_{\mathrm{opt}}-Y\right)+\left(D_{\mathrm{opt}}-Y\right)$$
where Y is the global average value for the particle size (in this case it should be 150 nm or near this value) and A_opt_, B_opt_, C_opt_ and D_opt_ are the best values of those four factors being studied.

According to previous works on the know-how and the variables involved in the process, four variables were considered for a preliminary Taguchi test: (A) TEOS addition speed, (B) reaction or maturation time, (C) temperature, and (D) surfactant concentration. All factors chosen were checked at three levels, the usual value employed till now and one above and another below it. In this case, according to the chosen parameters, the optimization matrix is a L9 matrix and requires the 9 different experiments as detailed in Table 1.

Figure 1 shows the effect of the four factors under study on the S/N ratio related to the particle size. The analysis of this figure indicates that the TEOS addition speed and the reaction or maturation time (factors A and B) are the most relevant variables affecting particle size. Firstly, a medium value of TEOS addition speed is the most convenient, while a high value of this variable produces a dramatic modification of particles sizes. Secondly, the highest value of the maturation time tested (120 min) seems to be the most adequate to reach the particle size of interest. In general, the optimal values of the variables or factors tested indicated by the Taguchi methodology here applied are A_opt_ = 0.5 mL⋅min^−1^, B_opt_ = 120 min, C_opt_ = 85 °C, and D_opt_ = 3 g⋅L^−1^ (the highest values). In these conditions, the optimal particle size as calculated with equation (3) is 162 nm.

Considering not only the particle size but also the porosity, which should be the expected one for MCM-41, the results of the experiments 1 and 2 provided oversized particles, while experiment 9 led to very small particles and yields. Regarding porosity, experiments 3, 4 and 8 did not show the typical diffraction results of MCM-41 MSNs (a very intense peak at 2θ = 2.4 and peaks of lower intensity at 2θ = 3.8 and 4.4, related to crystal faces 100, 110 and 200, respectively) and were discarded because desired MCM-41 morphology was not obtained (see Figure 2). Among the remaining experiments (5, 6 and 7), the conditions employed on experiment 7 were chosen as the best according to nanoparticles dispersion (relatively narrow, but includes particles whose diameter is bigger than 100 nm), size (164.2 nm according to DLS measurements and 209 nm according to TEM images; very similar to the optimal value suggested by selecting the most adequate levels for all factors according to Taguchi methodology) and porosity (MCM-41 like). Representative micrographs and size distribution histograms for experiments 5–7 could be found in Figure 3.

### 2.3. One by One Factor Analysis

Even with a 20% error in the particle size, the trends and values suggested by the Taguchi optimization study remain certain, as the S/R global average value changes from 3.7 to 4.2 (this value is represented by the dash lines in the graphs contained in Figure 1). However, in order to check the influence of some variables in wider operational intervals and to confirm the fluctuating influence of temperature on particle size, a study of the effect of each variable in a one-by-one basis was performed.

#### 2.3.1. Effect of TEOS/CTAB ratio

Other capital aspect on the synthesis of MCM-41 like MSNs is the reagents concentration ratio. The ratio between TEOS and CTAB plays an important role to modify the textural properties of MSNs, mainly as a consequence of the CTAB-mesophase template-driven synthesis. Indeed, it is easy to assume that if CTAB concentration decreases, it may get eventually depleted during the reaction as the CTAB remains trapped in-between the mesopores as the synthesis reaction progresses. Under these CTAB reduced concentrations, there were obtained MSNs nanoparticles with different porosity domains of between the core and the surface, as could be seen in Figure 4. The presence of such pore domains suggests that different interactions must take place between micelles and the forming silicates as the condensation progress. Additionally, it is also possible to postulate that, below a critical CTAB amount, the porosity shifted from longitudinal (surfactant-rich) to radial (surfactant-poor) distribution. Contrarily, when larger CTAB amounts were employed, this core-shell pore arrangement was not observed. Regardless of the pore distribution, it is also important to remark that similar sizes were obtained in all these cases, suggesting that TEOS amount is not the only parameter that dictates the resulting particle diameter distribution. These changing porous structures and the coalescence between particles could explain the differences in the particle’s diameter observed by DLS and TEM (Figure 4). Indeed, as, the amount TEOS amount is similar in both cases, the particle size increases must depend on the silane ratio to CTAB ratio.

#### 2.3.2. Effect of Temperature

In addition to the study of temperature according to the Taguchi design (see Table 1), a wider range has been studied at the low-volume scale because temperature is usually considered as a key variable regarding particle morphology. In this study, temperatures ranging from 50 to 90 °C were tested. Both the lower and upper limits were chosen considering the critical micelle-formation temperature and the boiling point of water, respectively. Although most of the previous works in the literature employ temperatures between 80 and 90 °C, some contributions proposed 60 °C as the best temperature for highly monodisperse MSNs [32,33]. Hence, herein a wide range of suitable temperatures from 50 to 90 °C in intervals of 10 °C was analyzed. Figure 5 shows the comparison of average particle diameters based on DLS and TEM measurements. Again, the size of nanoparticles estimated by DLS is slightly higher than that estimated by TEM measurements. Surprisingly, as shown in the micrographs contained in Figure 5, the highest temperature led to more amorphous and coalesced particles while temperatures from 50 to 80 °C provided smaller sizes with better morphology.

### 2.4. Volume-Based Scale-Up

Once the parameters were optimized for the 0.5 L scale, the synthesis of MCM-41 MSNs was carried out in a pilot plant cylindrical reactor of 5 L with 4 L working volume using a slightly lower temperature (80 °C), as this variable showed only a marginal effect on particle size, shape, and porous structure between 80 and 85 °C. Accordingly, several parameters and conditions were modified to the desired (×8) scale. Firstly, TEOS addition ratio was normalized in relation to the employed reaction volume, employing a value of 4 mL·min^−1^; as a first choice adequate for a wide interval of viscosities and for solid-liquid suspensions, a Rushton turbine was employed as overhead stirrer.

With this configuration, the MSNs obtained did not present the rounded shape of previous experiments, in particular, those typically obtained in the Taguchi set of runs, and were obtained as slightly elongated rods, with a length that exceeds the particle width in 1.5–2 times. In an effort to obtain more spherical particles, several overhead stirrers were tested: elephant ear, paddle, anchor, and Rushton turbine (samples B1 to B6 in Appendix A, Appendix A). In all runs performed, the resulting particles had diameters comprised between 180 and 380 nm by DLS or between 115 and 200 nm according to TEM images, but with no apparent relation to the impeller selected. After several syntheses, we assumed that this effect could not be avoided as it was consequence of two insurmountable aspects: (1) a great difference on the fluid dynamic conditions caused by the simultaneous modification of reaction vessel and stirrer, and (2) the difference in shear force onto the forming particles that occurs when the magnetic stirrer is replaced by an overhead stirrer. In view of these results, and for subsequent experiments, we decided to choose the Rushton turbine as reference impeller.

The agitation rate was varied in a short interval. The upper limit was set to avoid air mixing in the liquid phase, claimed to be a key variable in scale-up of silica-based microparticles [35]. Stirring speeds were set at 400 and 650 rpm and two runs were performed for each of these speeds. The results contained in Figure 6 indicate that this variable scarcely affects particle diameter and shape, if we consider error bars in TEM images. As usual, MSNs diameters estimated from TEM images are smaller than those from DLS measurements, as these later represent the hydrodynamic diameter, while the former is an average of all maximal cords taken from each angle between two opposite points in the perimeter (Feret diameter). If DLS diameters are considered, the nanoparticles are smaller at lower stirring speed, suggesting that an excessive stirring promotes coalescence in the pilot-plant 5 L reactor.

Finally, it was necessary to discard the previously seen thermal effect at lower temperatures than those used in the Taguchi DOE. Accordingly, and in line with the runs of the study of the effects of temperature alone, high scale runs were carried out at 60 °C. Under this condition, the average particle size moderately decreased, but, more importantly, the rod-shaped particles length was 1.1 to 1.5 higher than its width, so particles were more spherical.

#### 2.4.1. Effect of TEOS Addition Speed

The experimental results, after Taguchi optimization, seem to indicate that higher TEOS addition rates together with shorter curing times provide smaller nanoparticles. This is logical when considering that a faster TEOS hydrolysis may produce much more nucleation points and that, by the end of curing step, very little silica precursors would be available in the reaction media. Nevertheless, to confirm this assertion, several experiments were performed where TEOS addition speed was modified from 2 to ca: 11 mL⋅min^−1^, and even a fast addition at 290 mL⋅min^−1^ was performed. At addition rates higher than 11 mL⋅min^−1^, it was evident in micrographs that MSN particles formed conglomerates, so coalescence was a driven phenomenon under these conditions. In Figure 7, the results of experiments at low and medium addition rates are shown. In the experimental interval checked, the geometry of the particles was conserved: they were short rods with a length-to-width ratio of 1.3–1.6; however, the presence of conglomerates of particles is possible and can explain the different trend and value in particle size at increasing addition speed. For example, a close look at micrographs at addition rates of 4 mL⋅min^−1^ or higher indicates coalescence points that join some particles, though they are not too frequent. In this regard, only the slowest addition rate leads to equal values for particle diameter regardless of the analytical tool employed.

#### 2.4.2. Effect of Maturation Time

Reaction time is one of the variables that differs more between synthetic procedures reported in the literature. Along this line, the evolution of MSNs against increasing curing times was studied, though relatively high maturation times (up to 2 h) are common. To see the effect of maturation time and, in addition, to observe if it is possible to reduce it, MSNs were prepared under typical conditions taking aliquots at different times. Surprisingly, only 5 min after completing TEOS addition was enough to obtain nanoparticles with a full morphology and their aspect did not suffer relevant modifications at longer times, as is shown in Figure 8. Considering both DLS and TEM measurements, and their possible errors, a particle average diameter around 200 nm was obtained.

#### 2.4.3. Comparison of MSNs at Different Scales

As a final observation, we can compare the silica nanoparticles obtained at 60 °C in a round bottomed spherical flask with magnetic stirring and in the 5 L cylindrical reactor with overhead stirring provided by a Rushton turbine in terms of external and internal physical properties and chemical composition of the MCM-41 particles synthetized.

Regarding the particle size, at laboratory scale, the diameters obtained by DLS and estimated from TEM images were 248.4 ± 4.14 nm and 182 ± 54 nm, respectively, while for nanoparticles obtained at higher scale, they were 183.7 ± 2.36 nm (DLS) and 173 ± 55 nm (TEM). As for the shape, we can consider the aspect ratio and the circularity, which can be estimated with these equations:(4)$$\mathrm{Aspect}\;\mathrm{ratio}=\frac{\mathrm{Minimum}\;\mathrm{Feret}\;\mathrm{diameter}}{\mathrm{Maximum}\;\mathrm{Feret}\;\mathrm{diameter}}$$
(5)$$\mathrm{Circularity}=\frac{4\cdot \pi \cdot \mathrm{particle}\;\mathrm{area}}{\mathrm{particle}\;\mathrm{perimeter}}$$

We found that the nanoparticles obtained at laboratory scale average a circularity of 0.99 ± 0.009, suggesting a shape very similar to a sphere (see Representative TEM images section on Supplementary Material, pages S8 to S14, and Figure 5). While the MSNs obtained in the cylindrical pilot-plant reactor had an average circularity of 0.94 ± 0.028, which is slightly lower and corresponds to a short-rod like particle. Another common shape parameter is the aspect ratio, which is near 1 for MSNs obtained at low scale (0.90) and notably lower (0.77) for MSNs obtained at higher scale, indicating the difference in shape between both synthesis of nanoparticles.

Despite both laboratory and pilot-plant syntheses produce hexagonal prisms, it is relevant to remark that they show similarities in widths and only vary slightly on the lengths of the resulting rods. Curiously, this effect seems not to be related with the stirring speed, as the synthesis performed at 400 rpm and at 650 rpm show highly similar aspect ratios (0.771 vs 0.776) and circularities (0.95 and 0.94, respectively). This fact seems to indicate that the typical fluid dynamics of a turbine-stirred reactor is the governing factor on particles morphology. This effect is of importance as particle shape has a notable impact in the clearance of the MSNs from the body via renal or hepatobiliary routes. Moreover, shape has an impact on the particles distribution in the body [6] which affects their toxicity and bioavailability [42,43]; being rod shaped, MSNs are more bioavailable than spherical ones, although at the expense of having higher toxicity.

Similarity index (SI) values as proposed by Šoltys et al. [35] should be zero or near zero to indicate a perfect match between the property of the particle at the highest scale and the same property for the particle of reference (in a scale-up study, the particles obtained in the small reactor).
(6)$$\mathrm{Similarity}\;\mathrm{index}=\frac{P_{i}-P_{i0}}{P_{i0}}$$
being P_i_ the property value of the particle being tested (average value) and P_i0_ the value of the same property for the particles of reference. In this case, for MSN diameter SI values are 0.26 and 0 for DLS and TEM measurements, so, in this regard, as TEM permits a perfect observation of the individual nanoparticle, particle size similarity is perfect. For silica microparticles with a bimodal particle size distribution, volume-based scale-up ends up with SI values of 0.15 and 0.39, depending on the reference for an 8x scale-up (identical to the one used in this study), values very similar to the one computed in this study. As for particle shape, this geometrical similarity is also very reasonable: 0.14 for aspect ratio and 0.05 for circularity, even with the slight elongation of the rod-like nanoparticles observed in the cylindrical reactor.

Beyond the similarity in external properties of the MSNs, internal structures should be preserved as well as chemical composition. These parameters are compiled in Figure 9. For example, the average pore diameter at the smallest scale is 2.68 ± 0.42 nm, while this parameter is 2.96 ± 0.53 for MSNs obtained in the pilot-plant reactor (SI value of 0.104). The specific surfaces for both MSNs are 1254 and 1180 m^2^⋅g^−1^, again indicating that the internal structure is preserved in the scale-up process.

The same similarity is observed in the SAXS patterns of prepared nanoparticles of interest: the spectra show the typical long-range ordering diffraction pattern for MCM-41 mesoporous silica with the 100, 110, and 200 peaks being similar in all cases (for more information, refer to Supplementary Material).

## 3. Materials and Methods

### 3.1. Reagents and Equipment

Ammonium nitrate, tetraethyl orthosilicate (TEOS, different qualities), cetyl-trimethyl-ammonium bromide (CTAB) and NaOH were purchased from Sigma-Aldrich and used as received. Absolute ethanol was obtained from Panreac. All syntheses were accomplished using double distilled water (Younglin aquaMAX ultra) and carried out in either 1L rounded bottom flask employing 40 mm oval magnetic stirrers placed onto an IKA RCT basic hotplate stirrers or in a 5 L glass jacketed reactor fitted with a Heidolph Hei-TORQUE overhead stirrer. In all cases, the silica precursor (TEOS) was added with the aid of a New Era NE-300 syringe pump. pH values were measured with a Metrohm 827 pH lab apparatus.

### 3.2. Synthetic Procedure and Surfactant Removal

The typical synthesis method for MCM-41 mesoporous silica nanoparticles was achieved following a modified Stöber methodology in which 5 mL TEOS were dropwise added (0.25 mL·min^−1^) over a thermally stabilized solution (80 °C) composed by 1 g CTAB, 3.5 mL of 2M NaOH (14 mM) in 500 mL of double distilled water (resulting pH ≈ 11.8). Once the addition was concluded, the mixture was left to react for a certain time (up to 2 h) to allow full silica condensation. The obtained MSNs were isolated by centrifugation (10,000 rpm, 10 min) and repeated washings with 500 mL water (×2) and ethanol (×2) to remove excess reagents. The surfactant contained within the pores was removed under very mild conditions by washing the recovered nanoparticles with a slightly acidic ion−exchange solution (2×2 h reflux cycles with 250 mL of a 10 g·L^−1^ solution of NH_4_NO_3_ in 95:5 ethanol/water). Finally, the obtained particles were washed with EtOH (3×100 mL), dispersed in ethanol (50 mL) and stored in a refrigerator. For certain analysis prepared particles were dried under vacuum in an oven at 40 °C for 48 h. The absence of surfactant within the mesopores was assured by FTIR analysis of dried particles (Supplementary Material, pages S22-S23).

Particle syntheses at larger scale were carried out following similar procedures. Thus, TEOS (40 mL) was dropwise added over a thermally stabilized solution of CTAB (8 g), NaOH (2.23 g, pH ≈ 11.8) in water (4 L). In these experiments only a portion of the resulting colloid was taken for further isolation, surfactant extraction, and characterization.

### 3.3. Analytical Methods

Prepared MCM-41 mesoporous nanoparticles were analyzed employing available techniques. Size and morphology were visualized with the aid of JEOL JEM 1400 or 2100 Transmission Electron Microscopes (Tokyo, Japan) (analyzing an average number of 100 particles per run) and, in some cases, of a JEOL 6400 JSM Scanning Electron Microscope (Tokyo, Japan). X-ray diffraction (XRD) patterns were obtained with a Philips X-Pert Modular Powder Diffractometer (Eindhoven, The Netherlands) equipped with Cu K_α_ radiation source. Nitrogen adsorption isotherms were recorded in a Micromeritics (Norcross, GA, USA) 3Flex instrument and Fourier transform infrared (FTIR) spectra were obtained with a Nicolet Nexus (Thermo Scientific, Madison, WI, USA). 

Zeta potential and hydrodynamic diameters were recorded in an ethanol suspension employing a Zetasizer Nano ZS from Malvern Instruments (Malvern, UK) equipped with a 633 nm laser. Particles centrifugation and drying were accomplished with the aid of a Sorvall Legend XTR centrifuge (Thermo Scientific, Madison, WI, USA) and a Selecta Vaciotem-T vacuum oven (Abrera, Spain). Micrograph-based size analysis of chosen samples were performed employing ImageJ 1.52p (National Institute of Health, USA) and OriginLab® 2016 (Northampton, MA, USA) softwares.

Prepared ethanol colloids were analyzed by Dynamic Light Scattering (DLS) before surfactant removal. Given values for hydrodynamic diameter and surface potential are the average of 5 runs, except when indicated; and were performed upon dilution of the isolated colloid with absolute ethanol to a final concentration between 0.1 and 0.5 mg/mL (determined afterwards).

Nitrogen adsorption isotherms, and FTIR and Small-angle X-rays scattering (SAXS) spectra were measured employing extracted and dried samples. Isotherms were measured at 273 K after subjecting each sample to a degasification step at 125 °C for 24 h. Surface area was obtained by applying the BET method. Pore size distribution was determined by the BJH method from the desorption branch of the isotherm. SAXS analysis of the prepared MSNs was employed to discriminate if the typical long-range ordering diffraction pattern typical of the hexagonal honeycomb-like MCM-41 silica was present; to assure such textural property, the representative 100, 110, and 200 diffraction peaks must be present.

### 3.4. Identification Of Tunable Variables And Quality Criteria For Prepared Particles

The synthesis of MCM-41 MSNs according to the methodology developed by Stöber consists only of three reagents and a solvent. This solvent, water, is the most ecological, available, affordable, and safe solvent known, so it will be maintained in this scale-up study. Regarding the reagents, the great majority of the syntheses described in the literature use TEOS as silica source and CTAB as a template for the honeycomb-like micelle arrays, typical of the MCM-41 structure, so these will also be maintained. Considering the third reagent, the base responsible of promoting aqueous hydrolysis of TEOS, examples in the literature could be found that use ammonia, triethanolamine, or sodium hydroxide. Among them, according to our previous experience, NaOH is a very convenient reagent, due to its low-price, handling simplicity, and reproducible results at laboratory scale. Therefore, if the variability on reagents is very limited, the process optimization should be addressed towards modifying the physicochemical parameters governing the process. Among them, pH, temperature, agitation speed (mass transport), reaction time, and addition rate of TEOS seem to be easily modifiable parameters.

On the other hand, given that one of the goals is to scale-up the process to obtain nanoparticles with applicability in nanomedicine, there are some requirements that must be met. One of the most important is their size, which must be comprised between 50 and 300 nm, in order to prevent quick renal clearance and embolisms. Another important aspect are textural properties, as more convenient release patterns are obtained with particles showing higher pore order and narrower pore distribution (ca. 2.5 nm). Regarding their morphology, much research has demonstrated that rod-like morphologies produce higher cell uptakes [42,43], although it is not very significant in comparison to spherical particles. In summary, our goal is to prepare MSNs with high pore order and size in the range of 100–150 nm, with low size dispersion, to have a significant loading capacity.

With all the premises detailed above, the MCM-41 scale-up will be focused on which reaction conditions affect more the reaction outcome. The easiest-to-tune parameters that govern the process are the following: stirring speed, pH initial value, reaction time, reagents’ addition rate, and temperature. To determine the effect of each single parameter, in a first step, a Taguchi optimization method was accomplished; therefore, the associated matrix correlating such parameters was prepared. This optimization methodology is based on carrying out experiments (synthetic processes) with slightly modified parameters (below and above the original values) to determine if the reaction outcome improved or not, accompanied of a statistical analysis. Thus, the Taguchi method is a design-of experiments (DOE) methodology that is both factorial (includes factors or process variables and levels or variable values) and orthogonal (process variables affect the objective or dependent variable independently). This results in a much fractionated factorial DOE, so a minimum number of experiments should be performed for given numbers of factors and levels. Moreover, the signal-to-noise analysis (S/N) permits a simple, yet effective, observation of the effect of each factor in the objective function and to predict the best possible conditions in the studied experimental range [44].

## 4. Conclusions

The influence of the main operational conditions for the MCM-41 MSNs synthesis have been studied al laboratory scale employing a Taguchi experimental design: TEOS addition rate, nanoparticles maturation time, temperature, and CTAB concentration, showing that all them influence the average particle diameter, and, in the best conditions, this value can be set at 160–165 nm. Furthermore, a more in-depth study of the effect of temperature at laboratory scale showed that 50–60 °C suffice to reach this diameter, while reducing coalescence (very evident at 90 °C). Moreover, we have confirmed that to avoid an enlargement of the MSNs diameter and a textural change, the mass/volumetric ratio of CTAB to TEOS should be 1 g / 5 mL (a molar ratio TEOS/CTAB of 8.2). Therefore, runs at a higher scale (×8 = 4 L) were all performed at 60 °C and with a TEOS/CTAB molar ratio value of 8.2.

The pilot-scale synthesis method for MCM-41 mesoporous silica nanoparticles has been studied with what regards stirring agitation, type of stirrer, TEOS addition speed, and MSNs maturation time. Stirring agitation scarcely affect average diameter, with a Rushton turbine as best impeller for this synthesis. A very slow TEOS addition rate is the best way to ensure the inexistence of coalescence between nanoparticles, as shown by the comparison of DLS and TEM results. Finally, the time of maturation of the formed nanoparticles affects their size, which is more evident in TEM images: it reduces as the maturation proceeds.

An interesting aspect arisen upon the scale-up process is the variation of morphologies suffered by the MSNs. Indeed, the rounded hexagonal prisms obtained in the laboratory scale suffer from a slight elongation when prepared on a cylindrical reactor with overhead stirring. In our opinion, this effect is caused by the different fluid dynamics in each reactor and the different shear forces near the stirrer. Additional studies on this effect are being considered for future research. Nevertheless, despite this particularity, the strict limitations needed for the use of those MSN as drug delivery platforms are met: MSNs produced at large scale do maintain an adequate size (100–200 nm) and a highly ordered MCM-41 porosity suitable for drug delivery nanosystems.

The best operating conditions at pilot plant scale, employing a Rushton turbine as overhead stirrer, determined in this work are a temperature of 60 °C and slow addition of TEOS (2 mL·min^−1^). However, as previously commented, these conditions could be slightly modified without affecting significantly the product characteristic obtained, which provides an additional value in regard to forthcoming industrial scale.

## Figures and Tables

**Figure 1 ijms-21-07899-f001:**
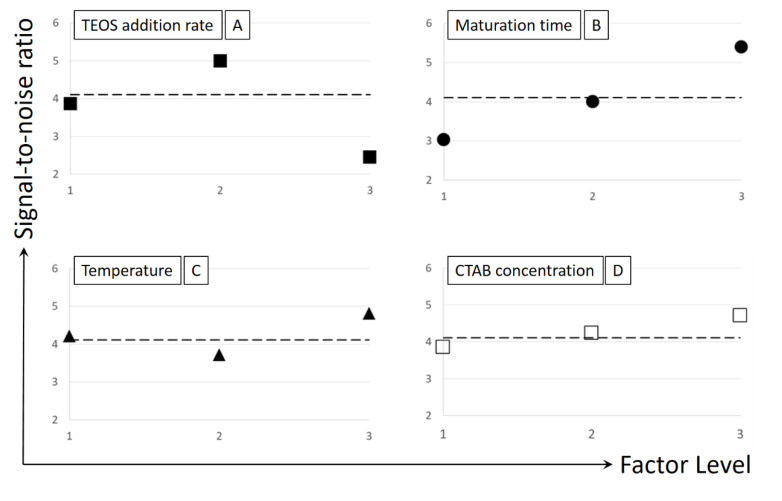
Effect of the variables or factors on the particle size according to the Taguchi L9 matrix: S/N ratio values versus level or value of each factor. Factors: ■ (**A**) TEOS addition rate; ● (**B**) Maturation time; ▲ (**C**) Temperature; ☐ (**D**) Surfactant or CTAB concentration.

**Figure 2 ijms-21-07899-f002:**
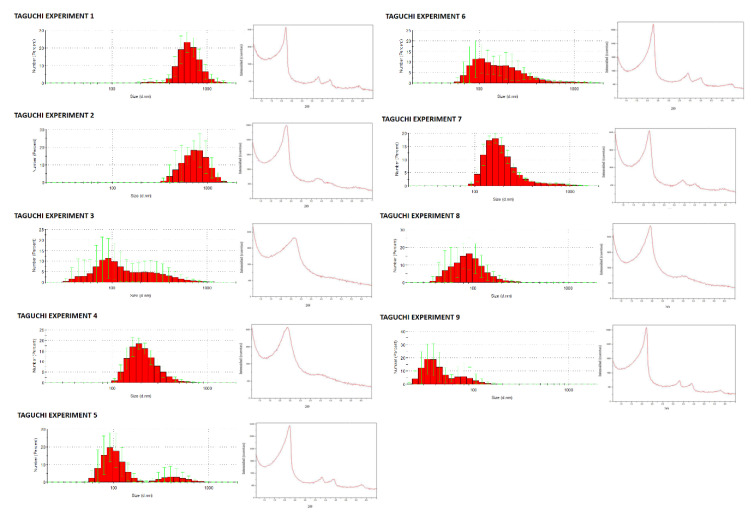
Dynamic Light Scattering (DLS) size distributions and Small Angle X-rays Scattering (SAXS) data for all the different Taguchi optimization experiments. Synthetic details are compiled in Table 1. DLS measurements were recorded in EtOH and are the average of 5 measurements. Graphic scales for SAXS range from 0 to 6.5 (2θ) and from 0 to 10^5^ counts following a potential curve (400x^2^). For DLS measurements *y* axis show percentages while the abscissa scale is logarithmic from 20 to 2000 nm.

**Figure 3 ijms-21-07899-f003:**
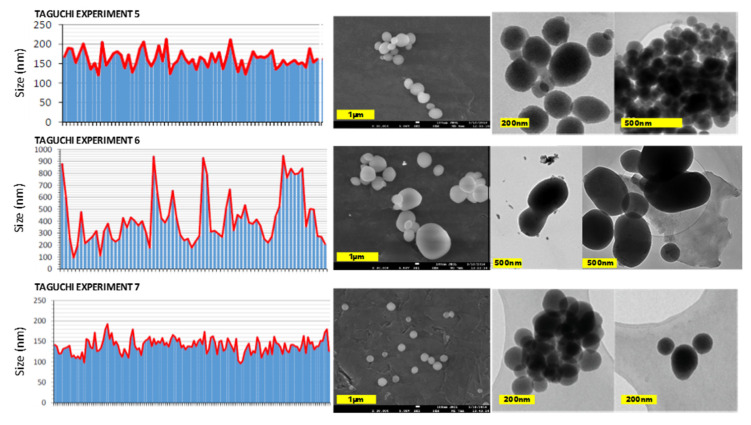
Micrograph-based size distribution, Scanning Electron Microscopy (SEM) and Transmission Electron Microscopy (TEM) images obtained for Taguchi experiments 5, 6 and 7.

**Figure 4 ijms-21-07899-f004:**
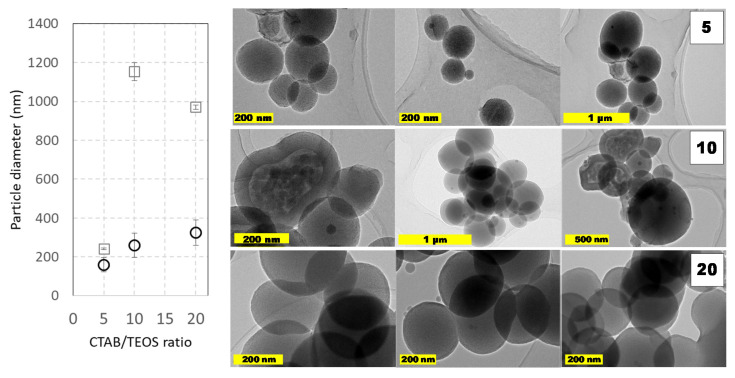
Particle size distributions and TEM images of prepared MCM-41 MSNs at varying TEOS/CTAB ratios. Squares (☐) are used for hydrodynamic diameters measured by DLS values and circles (◯) for diameters obtained from TEM micrographs. Experimental conditions: 1 L round-bottom flask reactor with 0.5 L working volume and magnetic stirring (800 rpm). Reactions were carried out at pH ≈ 11.8 (adjusted upon addition of 0.28 g NaOH) at 80 °C using TEOS (5 mL, 0.5 mL·min^−1^) and CTAB (1 g, 2 g⋅L^−1^) during a 60 min. Data shown correspond to samples D3, D5, and D7 in Supplementary material. Numbers on micrographs indicate the CTAB/TEOS ratio employed on each synthesis.

**Figure 5 ijms-21-07899-f005:**
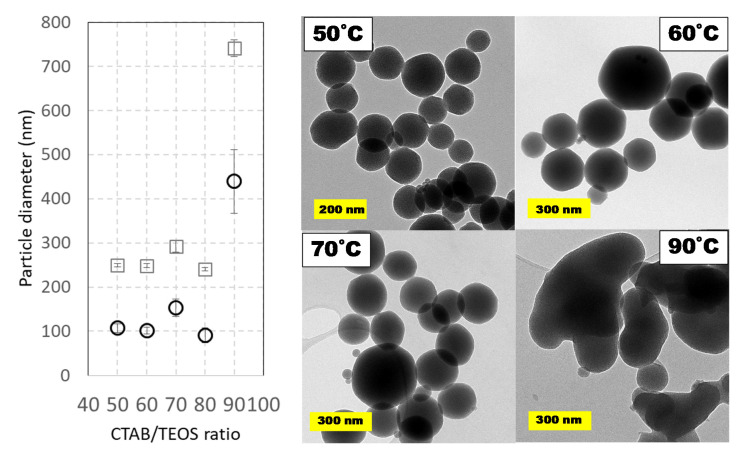
Particle size distributions and TEM images of prepared MCM-41 like MSNs at several temperatures from 50 to 90 °C. Squares (☐) are used for hydrodynamic diameters measured by DLS values and circles (◯) for diameters obtained from TEM micrographs. Experimental conditions: 1 L round-bottom flask reactor with 0.5 L working volume and magnetic stirring (800 rpm). Reactions were carried out at variable temperatures using TEOS (5 mL, 0.5 mL·min^−1^) and CTAB (1 g, 2 g⋅L^−1^) during a 60 min. Data shown correspond to samples C1 to C6 in Supplementary material.

**Figure 6 ijms-21-07899-f006:**
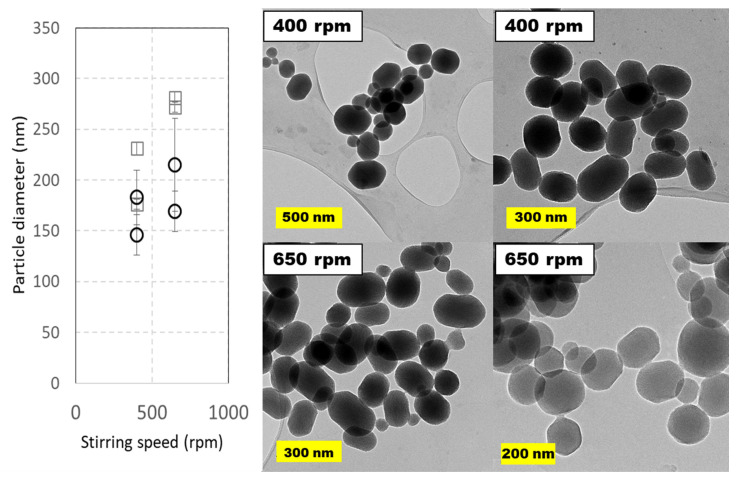
Particle size distributions and TEM images of MCM-41 MSNs prepared in pilot-plant scale under different stirring speeds (400 and 650 rpm). Squares (☐) are used for hydrodynamic diameters measured by DLS values and circles (◯) for diameters measured on TEM micrographs. Experimental conditions: 5 L jacketed cylindrical reactor with 4 L working volume with a Rushton turbine as overhead stirrer. Reactions were carried out at 60 °C using TEOS (40 mL, 4 mL·min^−1^) and CTAB (8 g, 2 g⋅L^−1^) during 60 min. Data shown correspond to samples 345.2, 345.4, 389.1 and 349.2 in Appendix A, Appendix A.

**Figure 7 ijms-21-07899-f007:**
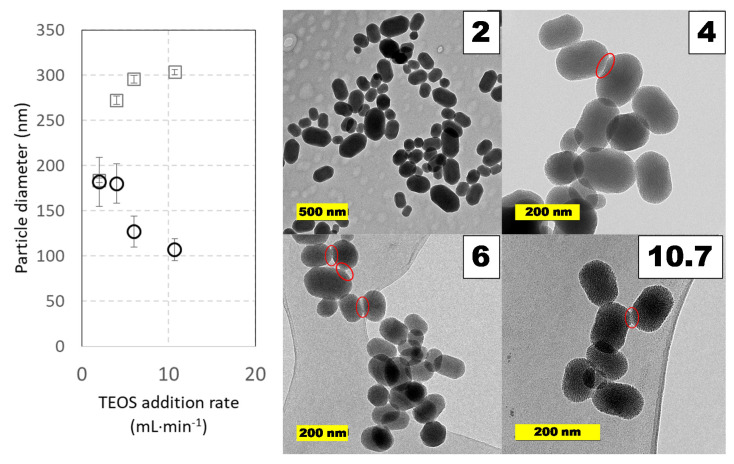
Particle size distributions and TEM images of MCM-41 MSNs prepared in pilot-plant scale under different reagent’s addition rates. Squares (☐) are used for hydrodynamic diameters measured by DLS values and circles (◯) for diameters measured on TEM micrographs. Experimental conditions: 5 L jacketed cylindrical reactor with 4 L working volume with a Rushton turbine as overhead stirrer (650 rpm). Reactions were carried out at 60ºC using TEOS (40 mL) and CTAB (8 g, 2 g⋅L^−1^) during 60 min at 650 rpm. Red elliptical contours show possible coalescence contacts between nanoparticles. Data shown correspond to samples 345.3, 349.2, 349.1, 345.7, and 345.6 in Appendix A of Appendix A.

**Figure 8 ijms-21-07899-f008:**
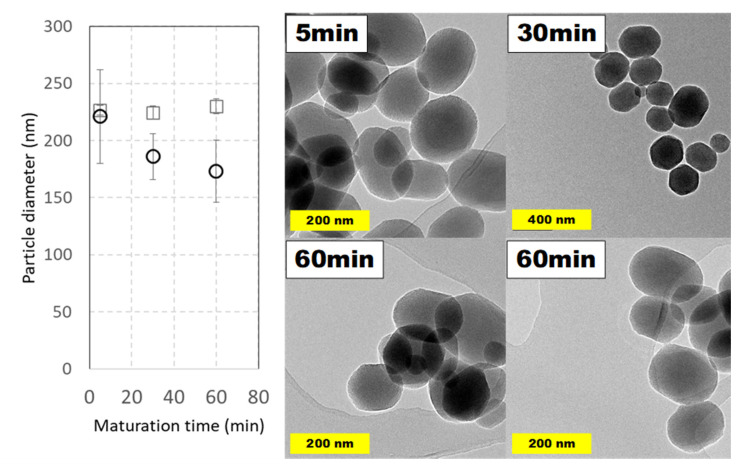
Particle size distributions and TEM images of MCM-41 MSNs prepared in pilot-plant scale at several reaction times. Squares (☐) are used for hydrodynamic diameters measured by DLS values and circles (◯) for diameters measured on TEM micrographs. Experimental conditions: 5 L jacketed cylindrical reactor with 4 L working volume with a Rushton turbine as overhead stirrer (650 rpm). Reactions were carried out at 60 °C using TEOS (40 mL, 4 mL·min^−1^) and CTAB (8 g, 2 g⋅L^−1^) during indicated times. Data shown correspond to samples 359.2 in Appendix A.

**Figure 9 ijms-21-07899-f009:**
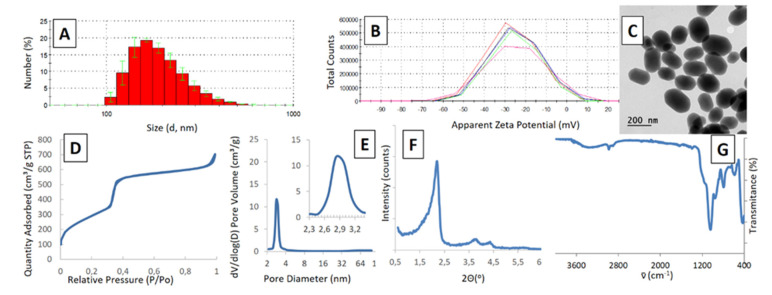
Characterization for scaled-up MCM-41 like MSNs Experimental conditions: 5 L jacketed cylindrical reactor with 4 L working volume with a Rushton turbine as overhead stirrer (650 rpm). Reaction carried out at 60ºC using TEOS (40 mL, 4 mL·min^−1^) and CTAB (8 g, 2 g⋅L^−1^) for 60 min. (**A**) Hydrodynamic size measured by DLS in EtOH, (**B**) Apparent Zeta Potential measured by DLS in EtOH, (**C**) TEM micrograph of resulting MSNs, (**D**) BET surface area calculated by N_2_ adsorption isotherm, (**E**) BHJ calculation of pore diameter from desorption isotherm (**F**) Small Angle X-ray diffraction spectra and (**G**) Infrared spectra of extracted nanoparticles. See Supplementary Material for full characterization of prepared nanoparticles.

**Table 1 ijms-21-07899-t001:** Experiments carried out according to Taguchi optimization method. Characters between brackets indicate values that are lower (1), equal (2) or higher (3) than those employed in typical runs. Experimental details: 1 L rounded bottom flask scale with magnetic stirring (800 rpm) employing 500 mL of double distilled water, 5 mL TEOS and 1 g of CTAB with the initial pH set to 11.8.

Run	TEOS Addition Speed (mL·min ^−1^)	Maturation Time (min)	Temperature (°C)	CTAB (g·L ^−1^)
**1**	0.33 (1)	30 (1)	75 (1)	1 (1)
**2**	0.33 (1)	60 (2)	80 (2)	2 (2)
**3**	0.33 (1)	120 (3)	85 (3)	3 (3)
**4**	0.5 (2)	30 (1)	80 (2)	3 (3)
**5**	0.5 (2)	60 (2)	85 (3)	1 (1)
**6**	0.5 (2)	120 (3)	75 (1)	2 (2)
**7**	1 (3)	30 (1)	85 (3)	2 (2)
**8**	1 (3)	60 (2)	75 (1)	3 (3)
**9**	1 (3)	120 (3)	80 (2)	1 (1)

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
