# Peer review of "Production of MCM-41 Nanoparticles with Control of Particle Size and Structural Properties: Optimizing Operational Conditions during Scale-Up"

_ijms, 2020, doi:10.3390/ijms21217899_

Round 1
Reviewer 1 Report
The research devoted to the scale-up procedure of MCM synthesis which is quite actual and interesting topic! There are several comments to the manuscript:
- Authors declare that " ... possible improvements [of resulting particles properties] are often accompanied by significant reductions in yield and worse texture properties" in Introduction (lines 79-80). References are required for such statements.
- Do the authors sure that the template was completely removed? How was it controlled?
- he authors used the Taguchi optimization method (lines 187-188) to determine the effect of each single parameter during synthesis. I suggest to provide a short method description or refs to make the manuscript more clearly.
- The values of the specific surface area should be accompanied by dispersion values.
- "...slightly rod-like shape of the particles..." (lines 333-335) I was not able to recognize rod-like shape of the particles on the provided Fig. Probably, any arrows of signs of Fig requried.
In addition, what is the reason of different results for hydrodynamic diameters measured by DLS and diameters obtained from TEM?
Reviewer 2 Report
This manuscript reports a systematic study of the formation of MCM-41 type mesoporous silica nanoparticles (MSN). Various reaction parameters such as pH, TEOS/CTAB ratio, temperature, stirring speed maturation time,… were studied. The effects of these reaction parameters on the texture and morphology of the formed MSN were studied via a Taguchi experimental design. The synthesis was finally up-scaled from the laboratory scale to the pilot scale (from a 500 mL flask to 5 L reactor). The aim was to synthesize MSN with controlled texture (homogeneous pore diameter) and particle size. The targeted particle size was 150 nm.
The variation of the main reaction parameters allowed the determination of the operational conditions for a robust and reproducible method of MSN synthesis on a large scale.
The manuscript is generally well written. The references give a good overview over the state-of-the-art. The experimental part is complete and describes well the synthesis and characterization procedures. The results and discussion part is nice.
Several specific comments:
Abstract, line 21: ‘to set these variables in 60°C and 8’; bad expression, please change.
Line 198: replace ‘hydroxyls’ by ‘silanol groups’
The quality of the charts in figure 2 is not sufficient.
Line 281: ‘unsuitable porosity (figure 2’): The porosity is not discussed in figure 2.
Line 335: as shown
Line 336: ‘led’ instead of ‘leaded’
Line 426 ff: round TEM and DLS values.
Some general remarks:
In my opinion, there is a problem with the ESI file. The ESI gives huge information about characterization of various materials (DLS, XRD, nitrogen sorption, TEM), but there is absolutely no link to the manuscript file. I cannot find the sample names in the manuscript, and the manuscript do not give any reference to the ESI data. The data in the ESI file should be limited to the information that is necessary for the discussion in the main manuscript.
The authors should give some information concerning the yield of the formed nanoparticles. They only give information about the quantity of used TEOS. Does the whole fraction of the TEOS yields in the formation of MSNs, or does also precipitation occur, at least partially?
Overall, the manuscript gives interesting information about the formation of silica MSNs. Up-scaling studies are important in view of a broader application of these objects. I therefore recommend publication after minor revision.
